# Quantum Dots Mediated Imaging and Phototherapy in Cancer Spheroid Models: State of the Art and Perspectives

**DOI:** 10.3390/pharmaceutics14102136

**Published:** 2022-10-08

**Authors:** Luca Dirheimer, Thomas Pons, Frédéric Marchal, Lina Bezdetnaya

**Affiliations:** 1Institut de Cancérologie de Lorraine, CRAN, CNRS UMR 7039, Université de Lorraine, 6 Avenue de Bourgogne, 54519 Vandœuvre-Lès-Nancy, France; 2Laboratoire de Physique et d’Étude des Matériaux (LPEM UMR 8213), ESPCI Paris, PSL University, CNRS, Sorbonne University (LPEM), 75005 Paris, France

**Keywords:** quantum dots, spheroids, imaging, photodynamic therapy, photothermal therapy, tumor targeting, nanoparticles

## Abstract

Quantum Dots (QDs) are fluorescent nanoparticles known for their exceptional optical properties, i.e., high fluorescence emission, photostability, narrow emission spectrum, and broad excitation wavelength. These properties make QDs an exciting choice for bioimaging applications, notably in cancer imaging. Challenges lie in their ability to specifically label targeted cells. Numerous studies have been carried out with QDs coupled to various ligands like peptides, antibodies, aptamers, etc., to achieve efficient targeting. Most studies were conducted in vitro with two-dimensional cell monolayers (*n* = 8902) before evolving towards more sophisticated models. Three-dimensional multicellular tumor models better recapitulate in vivo conditions by mimicking cell-to-cell and cell-matrix interactions. To date, only few studies (*n* = 34) were conducted in 3D in vitro models such as spheroids, whereas these models could better represent QDs behavior in tumors compared to monolayers. Thus, the purpose of this review is to present a state of the art on the studies conducted with Quantum Dots on spheroid models for imaging and phototherapy purposes.

## 1. Introduction

According to the last EU Commission Recommendation 2022/3689, nanomaterials are constituted of solid particles, which have one or more external dimensions in the size range 1 nm to 100 nm. Indeed, the size of nanoparticles typically ranges from some tens to hundreds of nanometers, while semiconductor nanoparticles termed Quantum Dots (QDs) have a size below 10 nm. In other words, a “quantum size regime” but not a nm size regime is characteristic for QDs [1]. Due to this size constraint, QDs tend to show quantum confinement effects when the size becomes less than the Bohr excitonic radius. This leads to multiplying applications since, for example, the band gap is tunable with respect to changes in particle size [2,3]. These optical properties, and particularly the ability to emit fluorescence in a much more stable manner than organic fluorophores [4], make them a prime candidate for dynamic imaging, particularly in the context of fluorescence-guided surgery (FGS). FGS is a real-time intraoperative technique in surgical oncology aiming to highlight tumors during surgical resection to facilitate tumor resection and achieve negative tumor margins [5,6,7]. Complete tumor resection with negative margins is critical for improving a patient’s prognosis [8]. In addition, QDs can emit fluorescence in the near infrared (NIR) range, which represents a real interest for FGS. Indeed, the NIR window is characterized by a deeper light penetration [9] and a reduced tissue autofluorescence [10], thus improving imaging conditions. To achieve tumor accumulation, different approaches are possible, as illustrated in Figure 1. In brief, delivery could be achieved through QDs properties (size, charge, surface modifications, etc.) or enhanced by active targeting. Alternately, QDs could be encapsulated in larger nanoparticles, themselves modified for active targeting.

Before considering clinical applications, answers must be provided about the ability of QDs to specifically target tumors and about QDs toxicity. To this end, many studies have been carried out in vitro with cancerous cell lines before evolving towards in vivo animal models. This trend can be highlighted when performing a literature search in the PubMed database. In July 2022, 8902 results were found for “(Quantum Dot) and (Cell)” and 4138 results for “(Quantum Dot) and (Animal)”. However, before moving towards animal models, most of these studies could have been carried out on 3D in vitro models such as spheroids. A spheroid is a 3D model which recapitulates the structure of small avascular tumors [11,12,13,14,15]. By forming a sphere of cancerous cells, certain parameters specific to small avascular tumors can be simulated, such as the existence of oxygen, nutrients, and pH gradients, and the stratification of cells as proliferative periphery, quiescent area, and necrotic core. Furthermore, this model can be complexified by forming cocultured/tricultured spheroids to better represent the impact of the tumor microenvironment and the stromal population. All this allows to evaluate the ability of QDs to penetrate and retain within a model mimicking small tumors and to assess QDs imaging potential (Figure 2).

However, only 48 studies were found when searching for “(Quantum Dot) and (Spheroid)”. Moreover, of these 48 results, 14 of them do not deal with spheroid models, thus bringing the real number of publications to 34. Therefore, spheroids are not yet widely used for QDs study, but they would perfectly fit the gap between 2D in vitro models and in vivo models. Furthermore, in a societal context moving towards the reduction of animal experimentation to strictly necessary studies, they would constitute a second selection step of studies before animal investigations. Thereby, the purpose of this review is to constitute a state of the art on QDs studies carried out in spheroid models, mainly addressing QDs as imaging agents, but also for fewer therapeutic applications, such as photosensitizer for photodynamic therapy (PDT) and photothermal agent for photothermal therapy (PTT). The potential of spheroid models to expand diagnostic and therapeutic investigations of QDs will also be presented.

## 2. Methods

### 2.1. Methods

This review was conducted through a systematic review according to the directions denoted by the Preferred Reporting Items for Systematic Reviews and Meta-Analysis (PRISMA). A comprehensive literature search of PubMed database was performed up to July 2022 using the following search terms: “Quantum Dot” and “Spheroid”.

### 2.2. Figures

Figures have been created with BioRender.com (accessed on 1 August 2022).

## 3. Quantum Dots

Initially discovered in 1980 by Alexei Ekimov, QDs, due to their exceptional optical properties, evolved and found numerous applications, including bioimaging [3]. QDs generally measure between 2 and 10 nm and have a spherical core-shell structure [16]. However, depending on coating thickness, QDs can reach up to 100 nm [17]. The core is composed of semiconductor materials such as CdSe, CdTe, InP, etc., and is surrounded by a protective shell, generally ZnS [18,19]. QDs display numerous advantages compared to organic dyes, especially high fluorescence emission, photostability, a narrow emission spectrum, and a broad absorption spectrum, leaving a large choice for the excitation wavelength [2]. Another advantage of QDs is their tunable size, shape, and material, which impacts the emission wavelength. Smaller QDs (2–3 nm) tend to emit in the blue wavelength while bigger ones (5–6 nm) emit in the longer wavelengths (orange, red, or infrared) [20]. This ability of QDs is explained at the electronic level. When an energy input occurs, electrons pass from a stable state to an excited state of higher energy, both separated by a band gap. The band gap energy increases as the QDs size decreases, resulting in a shift of absorption and fluorescence emission features towards shorter wavelengths, and conversely for larger QDs. These properties of QDs allow the obtaining of a strong signal-to-background ratio [21] and to realize multiple biomarkers staining simultaneously with chemically similar, but differently sized, QDs, with only a single excitation light source [22]. The major concern with QDs is their potential biological toxicity, especially for cadmium-based QDs. Indeed, QDs can be degraded in vivo, leading to the release of heavy metals. To overcome this, new cadmium-free QDs were designed, such as copper indium-based QDs [19], but also nonmetallic QDs like graphene QDs or carbon QDs [23].

The synthesis of metal chacogenide (MX; X = S, Se, Te) or pnictide (MX; X = P, As) semiconductor QDs is performed by a bottom-up approach, by reacting salt or molecular precursors of the QD material components together in a solvent, in the presence of capping molecular ligands. Optimization of theses syntheses has been and still remains a topic of intense research, in particular in the context of bio/medical applications where low toxicity, brightness, and photostable emission in the near infrared transparency range of biological tissues are sought. Syntheses in aqueous media are considered as a “greener” approach but usually display less versatility in the choice of materials. Most syntheses are performed in high boiling point organic solvents at temperatures of typically 200–300 °C in order to promote the formation of high purity nanocrystals. The presence of hydrophobic capping ligands ensures the colloidal stability and enables control over the size and shape of the nanocrystals [24]. In a second synthesis step, a high band gap semiconductor shell is often grown around the QD cores to improve the passivation of surface electronic states and optimize the brightness and chemical stability. Several review articles are available in the literature to provide further insights into the available materials, synthesis routes, and resulting optical properties [25,26,27]. The resulting QDs are then transferred to water, either by encapsulating them into amphiphilic lipids or polymers, or by exchanging the native hydrophobic surface ligands with new hydrophilic ones. The design of the surface chemistry is critical for bioapplications, as it controls interactions with biomolecules from the medium, and hence, ultimately, their interactions with cells and in vivo biodistribution [28].

On the other hand, carbon QDs can be synthesized by both top-down approaches (laser ablation, electrochemical methods) or by thermal treatment of molecular precursors such as citrate, carbohydrates, or polymers [29]. Their fluorescence arises from surface defects and is strongly dependent on surface modification and passivation with hydrophilic polymers. Depending on the synthesis route, surface treatment, and doping with metallic or nonmetallic heteroatoms, the fluorescence emission may be tuned throughout the visible and near-infrared spectrum [30]. These carbon nanomaterials are attracting increasing interest, but the chemical synthesis mechanisms still need to be better understood to improve their reproducibility and optimize their optical properties, and the origin of the photoluminescence remains under debate.

QDs are ideal candidates for many biological applications. An indicative but not exhaustive list of these applications could be immunoassays [31], cell labeling [32], molecular and cell tracking [33,34], and in vivo dynamic imaging [35]. For biological purposes QDs are generally coated with biocompatible polymers like polyethylene glycol (PEG) or sulfobetaine to reduce nonspecific interactions and increase circulation lifetime [36,37]. QDs are good candidates for dynamic imaging due to their higher photostability compared to organic dye, as highlighted when compared to routinely used Alexa 488 (Molecular Probes, Eugene, OR, USA), where signals from Alexa disappear after 60 s while occupying much more longer times for QDs [4].

Nanoparticles, including QDs, have evolved over three generations. First-generation NPs were not specific to targeted tissue and were quickly captured by the immune system, leading to high hepatosplenic accumulation and a short circulation time. Second-generation NPs introduced the notion of stealth. They are coated with polymer chains like PEG, allowing them to escape the immune system, increasing their water solubility, and reducing NP’s aggregation. However, they still lack specificity [38,39]. Nevertheless, in certain contexts like cancer imaging, these NPs can accumulate in the tumor due to the Enhanced Permeability and Retention (EPR) effect [40]. This phenomenon occurs in solid tumors with a leaky and anarchic vascular network and with little or no lymphatic drainage. Increased vascular permeability and a lack of lymphatic drainage allow NPs to accumulate in tumor areas. However, this effect is heterogeneous and can vary greatly between tumors, patients, and disease stage. Finally, third-generation NPs were functionalized with different ligands like protein, peptides, nucleic acids, etc., allowing them to perform active targeting [38,39]. However, in some cases these modifications lead to fluorescence quenching and thus should be carefully considered.

Among QDs, particular interest is set on near-infrared (NIR)-emitting QDs. Indeed, NIR light penetrates deeper into tissues, up to several centimeters, and is less absorbed [9]. Furthermore, tissue autofluorescence is reduced in the NIR window, especially between 650 and 950 nm, due to a reduced absorption coefficient from oxygenated and deoxygenated blood, but also from skin and fatty tissue [10]. Thereby, NIR QDs are very interesting in bioimaging, especially for cancer imaging.

## 4. Spheroid Models

Progress in drug screening requires physiologically accurate in vitro cell models, which best reproduce the in vivo situation. Cells cultured in 2D monolayers are a fundamental model for research. Their ease of use combined with a simple structure makes them a wise choice for any new study. However, this model does not reproduce certain parameters specific to tumors, and this is where multicellular tumor spheroids (MCTS) are introduced.

A spheroid is a 3D in vitro cell model used to mimic the structure of small avascular tumors. Several spheroid formation techniques have been developed [11,13,14,15,41], which can be roughly separated in two categories: scaffold-based and scaffold-free. In scaffold-based approaches, spheroids grow within a natural/synthetic matrix such as Matrigel^®^ (Corning Life Sciences, New York, NY, USA), collagen, agarose, etc. In scaffold-free methods, cell adhesion is prevented, promoting cell aggregation and thus spheroid formation. In the latter, different techniques have been developed: Liquid Overlay, where cells are cultivated in supports coated with non-adherent materials; Hanging-Drop, where drops of medium containing cells are suspended to promote cell aggregation under gravity; the Spinner technique, where cells are kept in suspension in agitated systems; and microfluidic systems.

The first advantage of spheroids is their ability to mimic the characteristics of small avascular tumors by forming a sphere of tumor cells. In this configuration, only part of cells, in the periphery, are directly in contact with a medium and thus with drug/NPs, while most cells are hidden within the spheroid, escaping direct contact with environment [15]. Furthermore, spheroids can simulate cellular heterogeneity as found in tumors [11,41]. This heterogeneity can be manifested by the presence of different cell types, but also by different cell states within the same population. Indeed, cells inside spheroids are generally divided into three layers: a proliferative periphery, a quiescent center, and a necrotic core. This stratification results from the presence of gradients within spheroids. Approaching the core, nutrient concentration decreases, as well as dioxygen, leading to hypoxic or even anoxic areas, while pH becomes more acidic. A known phenomenon responsible for pH acidification is the Warburg effect [12]. In response to hypoxia, cells promote an anaerobic metabolism and convert pyruvate into lactate to generate energy, acidifying the pH. The lack of oxygenation generally appears beyond 120 µm depth, limited by O_2_ diffusion [11]. This stratification is influenced by spheroid diameter, as small spheroids (<150 µm) may not exhibit marked stratification compared to larger spheroids (between 200–500 µm), while beyond 500 µm the necrosis area will be further increased [42]. In vivo, these gradients are due to abnormalities in tumor vascularization [43]. This stratification can influence treatment outcomes. For example, drugs targeting proliferative cells like paclitaxel are less efficient in a spheroid core, composed of quiescent and necrotic cells, compared to the periphery [44]. Another advantage of spheroids compared to 2D cultures is their ability to reproduce a more complex extracellular matrix (ECM), enriched with fibronectin, collagen, and laminin [45,46]. ECMs can influence cell behavior by modulating adhesion, proliferation, migration, and invasion [47]. It can also act as a barrier to nanoparticle delivery [48]. Most studies using spheroids are focused on a single cell type, while tumors are composed of a mixture of stromal and tumoral cells. Thus, spheroids consisting of a single cell population (homospheroids) may suffer from low cell diversity and incomplete ECM composition. To obtain a more accurate model, cocultured spheroids (heterospheroids) can be formed by mixing various cell types. Different types of cocultures are feasible, for example by coculturing cancer cells with fibroblasts/cancer associated fibroblasts [49], endothelial cells [50], and immune cells [51]; moreover, even triculture spheroids have been recently proposed [52]. In addition to cell types, other parameters can be modulated, such as the ratio between each cell population, but also the moment of seeding (simultaneous/deferred) [53]. These stroma-rich spheroids provide a more realistic tumor microenvironment, through cell-cell and cell-matrix interactions. These interactions, whether physical or through the release of soluble factors, will influence tumor cell responses, for example by inducing resistance mechanisms [54]. Some of the spheroids’ characteristics mentioned above, as well as their interest for QDs studies, are illustrated in the Figure 3.

Interactions between spheroids and inorganic nanoparticles have recently been reviewed by Henrique et al. [13]. Hence, only the main factors influencing NPs penetration through spheroids will be summarized here. NPs penetration is influenced by their size, shape, charge, and surface modifications [11,13,55]. Smaller NPs generally penetrate more successfully than their larger counterparts, as is the case with 20 nm polystyrene NPs, which penetrate more deeply in BxPC3 and PANC-1 spheroids than 500 nm ones [56]. The shape also plays a crucial role. Generally, rod-shaped NPs show higher penetration ability compared to spherical NPs. Indeed, HeLa spheroids were better labeled with short rod-shaped polystyrene NPs compared to long rod-shaped or spherical NPs [57]. However, further studies are needed to better understand the impact of NPs size and shape, since sometimes unexpected results were obtained, as was the case with low aspect ratio cylindrical NPs (100 × 325 nm) that were more effectively delivered to HEK spheroids compared to smaller cylindrical NPs (100 × 200 nm) or nanorods [58]. Conversely, large spherical NPs penetrated less than smaller ones. Concerning NPs charge, cationic NPs are generally adsorbed in the outer layers of spheroids, while the anionic ones penetrate more deeply [13,14]. This is notably the case in the study of Ma et al., in which negatively charged QDs penetrated more deeply than positively charged QDs in HeLa spheroids [59]. However, the higher adsorption of cationic NPs sometimes leads to better accumulation than anionic NPs, with the example of cationic dendrimers capable of higher labeling of KB spheroids compared to neutral/anionic dendrimers [60]. Finally, surface modifications are a determining factor for NPs penetration. Coating with PEG reduces aggregation and nonspecific adsorption of proteins, while increasing circulation lifetime [61]. Functionalization with various ligands such as peptides, proteins, antibodies, aptamers, and other molecules for active targeting purposes allow the specific targeting and increase of NPs delivery [62].

## 5. State of the Art: Quantum Dots Studies in Spheroid Models

Quantum Dots can be used as single nanoprobes or as parts of nanoparticular complexes. As a single nanoprobe, they are mainly used for imaging purposes. However, they can also find therapeutic applications such as photosensitizers for photodynamic therapy or photothermal agents for photothermal therapy. In nanoparticular complexes, they can be a part of theranostic projects, combining diagnostics and therapy, or be encapsulated to improve QDs biodistribution.

### 5.1. QDs as a Single Nanoprobe

#### 5.1.1. Imaging Applications

Given their optical properties, the main application of QDs is imaging. As mentioned previously, they can be used for various imaging applications such as immunoassays, molecular imaging, cell tracking, etc. The research works described in this review mainly focus on cancer targeting for imaging purposes, and notably some of them aimed to improve fluorescence-guided surgery (FGS). Here, spheroids are used to simulate small avascular tumors and/or metastasis and evaluate QDs ability to penetrate and label tumors, which is not possible on 2D models since some crucial parameters such as complex matrices, barrier effect, and cell stratification are missing. QDs studies on spheroid models can serve as the basis to improve FGS, a real-time intraoperative technique whose purpose is to highlight tumors, including tumor margins, during surgical resection [5,6,7]. Imaging agents must accumulate in tumors before being excited at specific wavelengths with subsequent signal acquisition by specialized equipment. Different types of equipment have been developed, such as cameras placed above the operation field, integration to surgical instruments, portable handheld devices, etc. Collected data will be transmitted to surgeons to facilitate tumor resection and surgical margin determination [63], which is a major prognosis factor for patients [8]. Moreover, FGS can be improved by using NIR imaging agents. As mentioned above, tissue autofluorescence is reduced in the NIR window between 650 and 950 nm [10], enhancing the contrast between the labeled and unlabeled areas. Most conducted FGS studies focused on organic fluorophores because they are biocompatible and easily eliminated [39]. Different types of fluorophores are proposed for FGS; some of them are already FDA-approved, such as indocyanine green (ICG), fluorescein, methylene blue (MB), and protoporphyrin IX (PpIX) generated from 5-aminominolevulinic acid (5-ALA), while others like IRDye 800 CW are undergoing clinical trials [6,64]. Among them, ICG and IRDye 800 CW emit fluorescence in the NIR region, around 800 nm. However, these fluorophores are not very stable over time, leading to fluorescence extinction, which is problematic during long-term surgical operations [3,65]. Moreover, the emission spectra of these molecules are generally broad while excitation spectra are narrow, which can generate technical constraints. Conversely, QDs are very stable over time, and have a broad excitation spectrum and a narrow emission spectrum, which make them very suitable candidates for long-term imaging.

##### Untargeted QDs

The interaction between QDs and spheroids was studied by Jarockyte et al. and compared with in vivo results. Negatively charged carboxyl-coated QDs 625 ITK^TM^ (CdSe/ZnS, 22 nm diameter) were used in NIH3T3 mouse embryonic fibroblasts, MCF-7, and MDA-MB-231 human breast cancer spheroids. They first demonstrated by z-stack confocal laser scanning microscopy (CLSM) that 8 nM QDs mainly labeled NIH3T3 spheroid periphery up to the 3rd–4th cell layer after 24 h incubation and spheroid size (150/800 µm) had no impact on QDs penetration. In terms of breast cancer spheroids, MDA-MB-231 were tightly packed, while MCF-7 spheroids formed loose aggregates. In both spheroids QDs were localized at the periphery. However, the relaxed structure of MCF-7 spheroids allowed deeper penetration. To predict QDs penetration, the authors elaborated a mathematical model based on diffusive transport. This model postulates that QDs penetrate up to 15 µm after 1 h incubation and 25 µm after 24 h, irrespective of spheroid size. Experimental results supported this finding. Tumor xenografts in CB17 SCID mice were further generated by injecting MCF-7 or MDA-MB-231 cells around the nipple and showed strong labeling of tumors 24 h after intra-tumoral injection of QDs (50 µL, 1 µM). Consistent with spheroids, labeling occurred on tumor periphery. Finally, they compared QDs accumulation in both models. In monolayers QDs labeled 80–95% of breast cancer cells, and in spheroids 60–70% of cells, while less than 30% of tumor cells were labeled in vivo. Thereby, this study confirms the benefit of spheroids compared to 2D monolayers as it allowed to reproduce more accurately the QDs penetration profile compared to in vivo results, although the number of labeled cells remains higher. The important conclusion is that spheroid structure, rather than size, impacted QDs penetration [66].

Comparison of QDs behavior between monolayers and HeLa spheroids has also been done by Ma et al. using Qdot 605 ITK^TM^ amino PEG and Qdot 605 ITK^TM^ carboxyl. Compared to monolayers, spheroid cells had longer cell doubling time and fewer cells were found in S phase. Spheroids were spherical and filopodia could be found around cells. Hematoxylin and eosin staining, as well as Live/Dead assay using Calcein AM and ethidium homodimer, revealed a cell stratification with a proliferative periphery, a quiescent region, and a necrotic core. In 500 µm diameter HeLa spheroids, positively charged PEG-QDs sparsely accumulated on the periphery while negatively charged carboxylated-QDs were able to penetrate more deeply and more homogeneously, up to 90 µm, whereas both QDs strongly labeled 2D monolayers. Hence, this study underlined the different behavior of QDs between monolayers and spheroids and highlighted a better penetration of negatively charged QDs [59].

To improve QDs delivery, Saulite et al. used mesenchymal stem cells (MSCs) as a vehicle to target breast cancer cells. Carboxyl-coated Qdot 655 ITK^TM^ (CdSe/ZnS) measuring 14.55 nm, with a −35.1 mV zeta potential were used. 2D MSCs were fully labeled with QDs before spheroid formation to evaluate QDs release from spheroids. MSCs spheroids retained 56% and 35% of QDs after 24 h and 72 h incubation, respectively. Afterwards, cocultured spheroids with ⅔ of MSCs pre-labeled with QDs and ⅓ of breast cancer cells MCF-7 or metastatic MDA-MB-231 were generated to evaluate QDs transfer. After 24 h, 18% of MCF-7 and 31% of MDA-MB-231 were labeled. The higher accumulation in MDA-MB-231 compared to MCF-7 cells was also observed when monospheroids were directly incubated with QDs. Finally, the authors showed that QDs accumulation was 1.5-fold higher in 2D compared to 3D for MCF-7, but surprisingly, accumulation was 6-fold higher in 3D than in 2D cells for MDA-MB-231. Thus, MSCs can be efficiently used as vehicles to deliver QDs to metastatic breast cancer. The authors could not provide a mechanistic explanation for the better QDs uptake in 3D vs. 2D metastatic cells due to the poor penetration of specific transport inhibitors (CPZ, EIPA, …) in spheroids [67].

##### Targeted QDs

To achieve specific tumor targeting, Przysiecka et al. used CuInS_2_/ZnS QDs capped with 3-mercaptopropionic acid (ZCIS@MPA) and conjugated with iRGD peptide (iRGD/QDs). ZCIS@MPA QDs had an average size of 3.6 nm and emitted fluorescence at 641 nm. The peptide conjugation increased the zeta potential from −21.5 mV to −18.9 mV. The aim of this peptide was to target αVβ3 and αVβ5 integrins with the RGD motif, leading to a proteolytic cleavage, allowing a cryptic Cend Rule motif (R/KXXR/K) to interact with neuropilin-1 and mediate QDs penetration. Spheroids were generated by the hanging drop technique using HeLa cervical cancer cells or MSU1.1 normal fibroblasts and incubated with QDs. HeLa spheroids were labeled more efficiently with iRGD/QDs than with ZCIS@MPA at both 1 and 24 h. Confocal analysis confirmed that iRGD/QDs penetrate uniformly, up to the spheroid center. In MSU1.1 spheroids, low staining was observed at 1 h irrespective of the type of QD, while strong staining with ZCIS@MPA QDs was observed at 24 h. Low αV expression on normal fibroblasts compared to HeLa cells could explain this result. Thus, specific cancer cells targeting was achieved with iRGD/QDs. WST-1 assays performed on cell monolayers demonstrated a higher cytotoxic effect of iRGD/QDs on HeLa cells than on MSU1.1 fibroblasts, decreasing HeLa viability to 60% at 20 nM, while reducing MSU1.1 viability to 60% at 1000 nM. Thus, iRGD/QDs allowed a specific targeting of cancer cells and had a low impact on healthy fibroblasts [68].

Whether in in vivo tumors or in in vitro spheroid models, the challenge associated with QDs is their poor penetration depth, constraining them to a peripheral accumulation. Based on the observation that QDs conjugated to cell-penetrating peptides are often retained on the plasma membrane or trapped in the endosome, Derivery et al. used cell-penetrating poly(disulfide)s (CPD) for efficient delivery of streptavidin-coated QDs (CdSe/ZnS) through thiol-mediated uptake [69]. Based on the improved CPD-mediated QDs delivery in 2D Drosophila S2 cells, Martinent et al. evolved towards 3D models. For this, streptavidin-coated QDs^TM^ 605 (CdSe/ZnS) were conjugated with CPD in HGM spheroids (HeLa modified cells) generated with low attachment plates. By confocal spinning disc microscopy, they showed that 10 nM CPD-QDs homogeneously labeled HGM spheroids after 6 h incubation. Thiol-mediated transport allowed QDs to penetrate up to the center of spheroids (350–400 µm diameter) by transcytosis, which is of great interest for deep tissue imaging. On the contrary, spheroids incubated with streptavidin-QDs were almost unstained [70].

Other than peptides, QDs can be coupled with a single-chain variable fragment (scFv), as done in the study of Bidlingmaier et al. in which M1-scFv targeting MCAM in mesothelioma was used. Specificity of the M1-scFV was confirmed by inducing MCAM expression in immortalized benign prostatic hyperplasia cells BPH-1, which led to cell labeling. Based on their previous work [46], they generated 150–200 µm diameter tumor spheroids from surgical specimens of mesotheliomas. Various analysis confirmed the presence of cancerous cells, macrophages, and a collagen-enriched matrix, in accordance with primary tumors. Furthermore, they demonstrated that treatment combining TRAIL (TNF-related apoptosis-inducing ligand) and cycloheximide induces total cell apoptosis in monolayers while spheroids were resistant to treatment. Returning to the work of Bidlingmaier et al.: M1-scFVs were coupled with NIR streptavidin Qdot 705^TM^ (CdSe/ZnS) and incubated for 4 h at 50 nM with a tumor spheroid. Confocal analysis revealed a strong labeling with M1-scFv in mesothelioma cells, stained with cytokeratin, while control scFV did not [71].

Pérez-Treviño et al. used anti-HER2 affibody (Aff)-targeted QDs for HER2 detection in breast cancer spheroids. Aff (anti-HER2/negative) were respectively conjugated to QDs (605/545), resulting in the formation of 30 nm particles. They first demonstrated by flow cytometry and confocal microscopy that anti-HER2 QDs preferentially labeled HER2+ cell monolayers while no difference appeared with neg-QDs between HER2+ and HER2− cells. Afterwards, 120 µm diameter spheroids were coincubated with both Aff-QDs (1:1 ratio) prior to performing confocal microscopy. HER2− spheroids were nonspecifically labeled since signals from both QDs were detected, while HER2+ spheroids were highly labeled with red fluorescence from anti-HER2 QDs 605. Permeabilization with triton further increased labeling. Later, ratiometric analysis (RMA_FI_) was performed to discriminate specific and aspecific labeling. Anti-HER2 QD RMA_FI_ values were higher in HER2+ spheroids compared to HER2− up to 30 µm depth, after which signals merged due to limited penetration. With permeabilization, RMA_FI_ values were higher throughout spheroid depth, and orthogonal views allowed anti-HER2 QDs detection up to 70 µm. By performing a mathematical removal of non-specific Aff-QDs signals, the authors improved image analysis and further quantified HER2 density, which was 23.5% in nonpermeabilized HER2+ spheroids (8% in HER2−) and 48.3% in permeabilized spheroids (9.2% in HER2−). The study ended by performing cocultured spheroids (HER2+/−) using both cell lines to better reproduce tumor heterogeneity. Some spheroid regions were more successfully labeled than others by anti-HER2 QDs, revealing HER2+ cell clusters. HER2 density was 30.8% after permeabilization, thus between HER− and HER+ spheroids [72].

Another work focusing on QDs for cancer imaging was carried out earlier by our group in which CdSe/CdS/ZnS QDs were coupled to folic acid (FA) in KB folic acid receptor α (FR-α)-positive cells. QDs were coated with zwitterionic ligands before conjugation with FA, which increased QDs hydrodynamic diameter from 6–7 nm to 18 nm and resulted in a 50% fluorescence quenching (λem = 608 nm). Cell viability was not altered in either KB (FR-α+) or A549 cells (FR-α−) after 24 h with 12.5–200 nM QDs. In monolayers specific labeling was achieved of KB cells with FA-QDs but not with FA-free QDs, while A549 cells (FR-α−) were not labeled. Then, 500 µm KB spheroids were formed by spinner flask techniques and incubated with QDs, leading to a higher labeling with FA-QDs compared to FA-free QDs. Preincubation with folate inhibited FA-QDs labeling, confirming specific targeting. QDs distribution throughout spheroids was similar between FA and FA-free QDs, reaching 100 µm depth, yet with a higher fluorescence intensity for FA-QDs indicating a higher tumor targeting with folate-rich QDs [73].

Chen et al. developed a microfluidic device to simulate the tumor microenvironment and the challenges associated with tumor delivery. HUVECs cells were seeded in the first channel to form a microvascular layer, the second channel was filled with basement membrane extract to reproduce perivascular ECM, and the third channel was composed of U-shaped microchambers for spheroid formation. Injection of BT549 triple-negative breast cancer (TNBC) or T47D non-TNBC cells into the device resulted in spheroid formation after 24 h, measuring 120–125 µm of diameter. The culture was further expanded up to 14 days, where spheroids reached 180–195 µm and exhibited more than 90% cell viability. Concerning NPs, the authors loaded doxorubicin (DOX) into carbon dots (CDs) coated with PEG and Folic acid. CDs-PEG-FA/DOX measured 12 nm in diameter and exhibited a time-dependent DOX release at pH 6.4. FRα expression was evaluated by Western Blot analysis and was found higher in BT549 than in T47D cells. Afterwards, CDs-PEG-FA/DOX were injected into the first channel of their system. After 3 h, the fluorescence levels were similar in both channels a and b, meaning that CDs efficiently traveled through the microvascular layer and perivascular ECM. After 12 and 48 h, CDs-PEG-FA/DOX penetrated in the periphery and core of BT549 spheroids, respectively. However, they were less accumulated in T47D spheroids, reaching periphery only after 48 h, in accordance with FRα expression. Finally, the authors used a 96-well plate adapted to contain the spheroid microchambers, allowing measurements with classical microplate readers. Cytotoxicity assays revealed an increased inhibition rate with CDs-PEG-FA/DOX in both cell lines compared to free DOX or CDs-PEG/DOX without FA. However, cytotoxicity was higher in BT549 cells. Thus, their microfluidic system allowed the evaluation of the penetration of modified CDs within spheroids surrounded by ECM and a vascular network [74].

In our recent study, NIR-emitting QDs (CuInZnSe/ZnS) were coupled with A20FMDV2 peptide (A20-QDs) for cancer imaging of head and neck cancer (HNSCC) spheroids. QDs were coated with sulfobetaine-block-imidazole polymers (SPP-QDs) and then coupled with A20FMDV2 peptide (NAVPNLRGDLQVLAQKVART) to target αVβ6 integrin, overexpressed in HNSCC. QDs had a radius of 13 nm and emitted fluorescence in NIR region at 739 nm. Using MTT assay, we showed that FaDu (pharynx cancer) and MeWo (cancer-associated fibroblasts) cell viability was not impaired after 24 h with 100 nM QDs. Then, FaDu spheroids and FaDu/MeWo (1:1 ratio) cocultured spheroids were generated using liquid overlay technique and reached 427 and 478 µm 5 days post-seeding, respectively. A20-QDs labeled spheroids more efficiently than SPP-QDs (25% vs. 3%) in both spheroid models. In cocultured spheroids, 32% of total FaDu cells were labeled against only 8% of MeWo cells. By z-stack confocal microscopy, A20-QDs penetration was set to 74 µm in FaDu spheroids and to 56 µm in FaDu/MeWo spheroids, confirming a peripheral accumulation. Finally, we performed the time-gated microscopy by introducing a 40 ns delay between excitation and fluorescence detection, which eliminated autofluorescence and improved QDs signal detection [32].

Another work aiming to improve FGS was led by Asha Krishnan et al. in which authors designed an NIR nanoprobe for prostate cancer imaging. Pegylated Qdot 655 ITK^TM^ aminos (CdSe/ZnS) were conjugated with prostate-specific membrane antigen (PSMA) to target LNCaP prostate cancer cells. PMSA-QDs reached 34.2 nm diameter and −1.3 mV zeta potential. No alteration in cell viability was evidenced after 3 h with 1–500 nM PMSA-QDs on different cell lines. After demonstrating PSMA-QDs labeling abilities in LNCaP monolayers, the authors further developed 1200 µm diameter LNCaP spheroids with a 450 µm thickness using a chip-based system consisting of a Petri dish coated with agarose and punctuated with microwells of approximately 2400 µm in which cells were cultured. After 2 h incubation with 50 nM PSMA-QDs, spheroids were intensely labeled, with equal distribution between core and periphery and a strong labeling over a depth of 100 µm from the top and bottom of spheroids. On the contrary, unconjugated QDs or blockages of PSMA receptors resulted in poor labeling [75].

As seen above and as summarized in Table 1, QDs investigations carried out in spheroid models are of great interest because they provide data about QDs penetration and distribution within a sphere mimicking small avascular tumors. Most of the works described previously highlighted a partial accumulation of QDs, mainly on spheroid periphery, while labeling appeared intense in monolayers. When spheroids were not fully labeled, penetration values of QDs ranged from 15 µm to 100 µm. Thus, spheroid structure, and notably, the harder cell accessibility, significantly impact QDs behavior, allowing to obtain closer results to those obtained in vivo where tumors are not completely labeled. As pointed out by Jarockyte et al., NIH-3T3 spheroid size had no impact on QDs penetration depth. Their mathematical model showed that only incubation time increased QDs penetration. However, the spheroid structure, influenced by the cell type, had a notable impact on QDs. Indeed, MCF-7 loose spheroids allowed deeper QDs accumulation than tight MDA-MB-231 spheroids [66]. Another parameter influencing nanoparticles behavior, including QDs, is their charge. Generally, cationic NPs are quickly adsorbed by cells, which limits their diffusion, while the anionic ones can penetrate deeper [14]. Thus, most of the QDs used previously are anionic. The impact of QDs charge has been demonstrated in the work of Ma et al. Although cell labeling was similar in 2D, positively charged QDs only partially accumulated in spheroids, while negatively charged ones were able to penetrate more deeply and more homogeneously [59]. Considering a lack of penetration within spheroids, many QDs modifications have been investigated, especially for active targeting. Various systems have been developed, such as coupling to peptides, antibodies, etc. Among them, the work of Martinent et al. has shown that coupling with CPD (cell-penetrating poly disulfides) based on thiol uptake allowed full spheroid penetration by these targeted QDs [70]. Other attractive qualities of spheroids include the possibility to perform coculture with different cell types, allowing to carry out selectivity studies and to evaluate the impact of another cell population on QDs penetration, as it can promote it or act as a barrier. As seen in the study of Yakavets et al., QDs penetration was reduced from 74 to 56 µm when the spheroid evolved from monoculture (FaDu) to coculture (FaDu/cancer associated fibroblasts). In the same work, it was demonstrated that QDs preferentially labeled cancer cells rather than fibroblasts [32]. Similarly, in breast cancer spheroids cocultures composed of HER2+ and HER− cells, only HER2+ cells were labeled with anti-HER2 affibody-conjugated QDs [72]. Spheroids allow for reproducing a more complex extracellular matrix, as seen in the study of Bidlingmaier et al., in which spheroids established from mesothelioma recapitulate certain parameters of the original tumor (presence of cancerous cells, macrophages, a collagen rich matrix) and in which QDs specifically targeted cancer cells through an active targeting approach [71]. Different cell types also make it possible to study intercellular interactions, as done in the study of Saulite et al., in which QDs transfer from mesenchymal stem cells to cancer cells was demonstrated in cocultured spheroids [67]. Other systems were used, such as the microfluidic device in Chen et al. where Carbon Dots (CDs) needed to go through a vascular layer and a perivascular ECM to reach spheroids and finally label them [74]. Thus, these 3D studies allowed for greater knowledge about the ability to penetrate and label tumors.

#### 5.1.2. QDs for PDT and PTT

Although imaging is the main application of QDs, they can find other applications such as photosensitizers (PS) for photodynamic therapy (PDT) or photothermal agents (PA) for photothermal therapy (PTT). PDT is based on PS accumulation in the tumor, before being excited by specific wavelengths to produce reactive oxygen species (ROS) in the presence of oxygen. This reaction leads to tumor cells apoptosis/necrosis, disrupting of the tumor vascular network, and stimulation of the antitumor immune response [76,77,78,79]. Briefly, the reaction as depicted by the Jablonski diagram [80] consists of exciting PS electrons, which will pass from a ground state (S_0_) to an excited state (S_1_). Then, intersystem crossovers will lead to a triplet state (T_1_), with a longer lifetime (µs) compared to S_1_ state (ns), allowing electrons to interact with their environment by two different reactions. A Type I reaction consists of interactions between PS and the surrounding molecules, forming free radicals which will interact with oxygen, generating ROS [76,77,78,79]. The Type II reaction directly involves energy transfer to oxygen, generating ROS and notably ^1^O_2_. The challenge with PS is to achieve tumor-specific delivery, deep penetration, and biocompatibility in the dark. The excitation wavelength also plays a critical role since light must penetrate tissues until reaching PS. NIR light is very interesting from this point of view, since as we mentioned above, it penetrates more deeply across the tissues. Several PSs have already been approved, such as 5-Aminolevulinic acid and methyl aminolaevulinate (Ameluz^®^, Biofrontera, Leverkusen, Germany, AlaCare^®^, Photonamic, Pinneberg, Germany, Levulan^®^, DUSA Pharmaceuticals, Wilmington, NC, USA, Metvix^®^, Galderma, Paris, France, etc.); all of these are prodrugs of PpIX; porfimer sodium (Photofrin^®^, Concordia Laboratories, Oakville, Canada), temoporfin (Foscan^®^ Biolitec Pharma Ltd., Dublin, Ireland), and others. QDs as PSs are limited by their low ^1^O_2_ quantum yield and therefore are rather used as energy donors for other PS [81,82,83]. However, much better results could be obtained with carbon-based QDs such as Carbon QDs (CQDs) or graphene QDs (GQDs) [84,85,86].

PTT is based on photothermal agent (PA) causing an increase in local temperature after irradiation, preferably in the NIR range, leading to cell death [87,88]. Similarly to PDT, PA electrons are excited from S_0_ to S_1_ state, but then return to ground state by non-radiative vibrational relaxation, which generates heat. Above 42 °C, cell death occurs by apoptosis or necrosis. The clinical application of PTT is less advanced than PDT, but various PA have already been investigated, such as indocyanine green, gold nanoparticles, carbon nanomaterials, etc. The ability to produce heat will depend on the PA nature. Generally, organic PA have lower photothermal conversion but are biocompatible and biodegradable, while inorganic PA can achieve high photothermal yield but are less degradable and can be retained for long times. QDs as PS or PA have been subjects of in vitro investigations, and particularly GQDs and CQDs, which have interesting properties for PDT and PTT. A basic representation of the QDs-mediated concept of PDT and PTT anticancer therapies is displayed in Figure 4.

The studies carried out in 3D models to simulate QDs-mediated tumor destruction during PDT/PTT are sparse (Table 2). Carboxylated graphene QDs (GQDs) were used by Perini et al. as a part of the improvement of the INSIDIA 2.0 tool. This open-source ImageJ macro was designed to perform multiparametric analysis of spheroids, including spheroid area, perimeter, circularity, solidity, and entropy, without operator bias. GQDs measuring 0.8 nm were used as photothermal agents for PTT on U87 glioblastoma and PANC-1 pancreatic cancer spheroids. PTT was performed under NIR irradiation at 808 nm for 5 min (0.8 cm diameter spot, power density of 6 W/cm^2^) and increased temperature up to 42 °C. U87 spheroids were treated for 14 days with GQDs (200 µg/mL), doxorubicin (DOX, 1 µM), or both, with or without PTT, and pictures were analyzed by INSIDIA 2.0. DOX alone/with GQDs reducedspheroid area by 30%, which was further decreased up to 70% when performing PTT. In addition, there was a decrease in circularity, entropy, and core area. PANC-1 spheroids undergo the same protocol, except DOX was replaced by 5-Fluorouracil (5-FU, 100 µM). PTT increased spheroid area and entropy, while core area was reduced. Here, two types of spheroid’s destruction were analyzed. U87 spheroids become smaller, which is correlated with a decrease of global area, core area, and entropy, while PANC-1 spheroids start disintegrating and become loose, increasing global area and entropy but reducing core area. All these underline the interest of a multiparametric analysis for NPs/drug screening on spheroids, and the ability of GQDs to destroy various types of spheroids by PTT [89].

Li et al. synthesized GQDs to perform both imaging and PDT on HepG2 hepatocellular carcinoma. The GQDs were enriched with 1.43% fluorine (F-GQDs) to improve PDT efficiency and compared to nonfluorinated GQDs. The F-GQDs displayed a size of 2.1 nm, with fluorescence emission around 500 nm. The fluorescence quantum yield was 13.71% for F-GQDs and 5.8% for GQDs. The F-GQDs were biocompatible, as cell viability remains at 85% even at 300 µg/mL for 24 h. The authors incubated F-GQDs (200 µg/mL, 6 h) with 400–500 µm HepG2 spheroids and showed by CLSM a brighter labeling with F-GQDs compared to GQDs. Under 400–700 nm LED light irradiation (40 mW/cm^2^, 12 min), they showed a time-dependent ^1^O_2_ production with F-GQDs while a weak production was observed with GQDs. ^1^O_2_ quantum yield was 0.49 for F-GQDs and 0.24 for GQDs. In monolayers, PDT with F-GQDs reduced HepG2 cell viability up to 35% while it remained at 70% with GQDs. Similarly, in spheroid models PDT more drastically reduced spheroid diameter compared to GQDs, destroying them almost entirely. Thus, these fluorine-enriched GQDs demonstrated promising potential for combined use in imaging and PDT [90].

Another work of this team realized by Wang et al. focused on copper-doped carbon dots (CU-CDs) for bioimaging and PDT. By the pyrolysis method, they synthesized 2.8 nm CDs enriched with copper to improve their PDT potential. Fluorescence emission was maximum at 410 nm after excitation at 280 nm. In both HeLa monolayers and SH-SY5Y (human neuroblastoma) spheroids measuring 300–400 µm, CLSM results showed stronger labeling with 0.15 mg/mL CU-CDs than with CDs. Labeling was also brighter at 24 h compared to 6 h. After irradiation (400–700 nm, 40 mW/cm^2^), the ^1^O_2_ quantum yields were 0.36 for CU-CDs and 0.15 for CDs. Using SH-SY5Y spheroids, incubation with 0.15 mg/mL CU-CDs in the dark inhibited spheroid growth compared to control condition, but substantial inhibition was observed when performing PDT. This study confirmed the benefit of copper addition in CDs to improve PDT efficiency, destroying cancerous cells [91].

A multimodal NP combining imaging, PDT, and chemotherapy have been designed by Hashemkhani et al. for colorectal cancers treatment. Ag_2_S QDs emitting at 830 nm were electrostatically loaded with 5-ALA and anti-EGFR antibody cetuximab (Cet) was added. QDs uptake was linked to EGFR expression, as high EGFR-expressing SW480 cells were better labeled by Cet-QDs than low-expressing HT29 cells. The conversion of 5-ALA into PpIX was higher in SW480 than HT29 cells and was enhanced in both cell line with ALA-loaded QDs. Cet conjugation further increases PpIX production in both SW480 monolayers and spheroids. Similarly, ROS production was enhanced with Cet-QDs after irradiation with blue (420 nm, 5 min, 2.1 J/cm^2^) or red light (640 nm, 1 min, 15.5 J/cm^2^). When performing PDT with blue light in SW480 monolayers (after 4 h incubation), cell viability remained at 80% with ALA alone while decreasing to 43% with ALA-loaded QDs-Cet and 38% when fluorouracile was loaded for combined therapy. In comparison, cytotoxicity was less pronounced in HT29 and Cet conjugation had no noticeable effect. Red light further increased PDT efficiency, reducing cell viability to 10% with combined therapy in both cell lines. Finally, PDT was performed with blue light in SW480 and HT29 spheroids after 24 h incubation. After PDT, SW480 viability decreased up to 59% with ALA alone, 28% with ALA-QDs-Cet, and further with combined therapy. Surprisingly, HT-29 spheroid viability decreased up to 42% with ALA-loaded QDs-Cet, while it remained at 65–70% with ALA or ALA-loaded QDs [92].

Studies focused on QDs for PDT/PTT applications are not yet widespread. Therefore, only few publications were expected on this issue. As summarized in Table 2 and illustrated in Figure 5, spheroids can be used to simulate the impact of PDT/PTT on small avascular tumors. Indeed, the work of Hashemkhani et al. demonstrated that spheroids were more resistant to PDT than cell monolayers [92]. PDT or PTT, alone or in combination with chemotherapy significantly impacted spheroids, causing major damages and cell death. Depending on the spheroid type, its destruction occurred in different ways. HepG2 and SH-SY5Y spheroids had a significantly reduced diameter after PDT [90,91]. In the work of Perini et al., U87 spheroids decreased in volume, while PANC-1 spheroids became larger, but less dense, as a result of progressive cell disaggregation after PTT. In addition, their study highlighted the potential of the INSIDIA 2.0 macro as a tool allowing a quantified analysis of different parameters specific to spheroids, such as their total area, core area, entropy, etc., helping to guide their evolution in response to the treatment [89]. Regarding PDT, study on spheroid models is essential. Indeed, PDT requires the presence of oxygen to destroy cancer cells. However, many solid tumors are hypoxic in the core, thus hindering PDT efficiency [93]. Therefore, the core of spheroids, which is characterized by hypoxic or even anoxic areas, can perfectly mimic oxygen-deficient situations. The hypoxia characterizing spheroids had notably been demonstrated using QDs [94]. Amphiphilic polyethyleneimine derivatives (amPEIs) encapsulating QDs and Ru(dpp)32+ dyes were used to perform ratiometric oxygen sensing. QDs emission was independent of oxygen content while Ru(dpp)32+ dye emission was quenched in the presence of oxygen. Ratiometric photoluminescence spectrum values were used to determine oxygen concentrations in HCT116 spheroid, which were 1.33%, 8.42%, and 19.9% in spheroid core, middle and periphery, respectively. Notably, in the two studies using PDT (Li et al.; Wang et al.), spheroids were almost completely destroyed, although small clusters still remained, possibly corresponding to the most central part of the spheroids [90,91].

#### 5.1.3. QDs Toxicity

The main trouble hindering QDs use in clinics is their potential long-term toxicity. Many studies have been carried out to evaluate QDs toxicity, overwhelmingly with 2D in vitro models and in vivo models. The papers dealing with QDs toxicity assessed in spheroid models are sparse. QDs toxicity is linked to the release of ions and heavy metals like cadmium, especially when QDs are degraded, causing oxidative stress through REDOX reactions [65]. In addition, their nanometric size makes them likely to interact with many intra or extracellular components, disrupting the overall function of cells. Numerous factors can influence QDs toxicity. Thus, a meta-analysis was reported in 307 publications to establish correlations between 24 parameters and cadmium-based QDs toxicity [18]. Toxicity was influenced by QDs properties (diameter, surface modification, surface ligands, shell) and incubation parameters (concentration and exposure time), but also by the type of toxicity assay used. The chemical composition of QDs also impacts their toxicity, as replacing cadmium QDs (CdSe/ZnS) with indium-based QDs (InP/ZnS) significantly reduced QDs toxicity in both in vitro and in vivo *Drosophila melanogaster* models [95]. In the same way, in vitro, indium-based QDs have been shown to be much less toxic than their cadmium-based counterparts, as evidenced on MRC-5 fibroblasts where MTT results after 48 h concluded on an IC_50_ of respectively 75.6 nM and 8.4 nM. Moreover, In-QDs had no hemolytic effect compared to Cd-QDs which induced 75% hemolysis at 100 nM. In vivo, subcutaneous injection of 20 pmol of CuInS_2_/ZnS QDs in mice allowed to visualize sentinel lymph node but traces of indium remained in spleen and liver three months after injection [96]. QDs toxicity does not only depend on their chemical composition but also on their surface modifications as highlighted by MTT assay after 48 h with pegylated or nonpegylated Ag_2_Se. Both CAL-27 (tongue cancer) and HaCaT (human immortalized noncancerous keratinocyte) cells displayed 80–90% cell viability at 200 nM with PEG Ag_2_Se, while decreasing to approximately 55% with non-pegylated QDs. However, injection of both QDs (100 µL, 5.10^−6^ M) into Balb/c mice vein tail had no impact on body weight, which continued to increase over 61 days, emphasizing good physical state [97]. Studies with Cd-QDs have been conducted in various in vivo models such as zebrafish *Danio rerio* larvae, *C. elegans*, and *Bombyx mori* larvae and demonstrated signs of toxicity [2,98,99,100]. A pilot study aiming to evaluate QDs in vivo toxicity in nonhuman primates was carried out on six *Rhesus macaques*. QDs (CdSe/CdS/ZnS) were encapsulated into phospholipid micelles and injected intravenously (25 mg/kg). Blood and biochemical markers were monitored for 90 days, followed by histological analysis of various organs and no signs of toxicity were evidenced, although a significant amount of cadmium was retained in the liver, spleen, and kidneys. Two of the six animals were followed for a year and remained healthy [101]. Thereby, QDs toxicity assessment is difficult, as toxicity is influenced by QDs composition, experimental parameters such as toxicity assay are involved, and results diverge depending on the study and the model used.

Ulusoy et al. assessed QDs toxicity in human adipose-derived mesenchymal stem cells (hAD-MSCs) isolated from patients’ subcutaneous adipose tissues. The authors synthesized carboxyl-functionalized CdTe/CdS/ZnS emitting at 670 nm, with a fluorescence quantum yield of 64%. They generated 312 µm hAD-MSCs spheroids and incubated them for 24 h with QDs. Under these experimental conditions, no morphological alterations were detected up to 300 µg/mL, while at 600–1200 µg/mL cells become more granular and spheroids size increases as they become looser. Then, QDs were incubated with cells before spheroid formation. Again, no toxicity was found up to 300 µg/mL, while spheroid formation was impaired at 600–1200 µg/mL, resulting in small aggregates formation. Hence, QDs were deleterious only at highest concentration when cells were not protected by the barrier effect of spheroid. To precisely assess cell viability, an ATP assay was performed. IC_50_ were similar whether QDs were incubated before (190 ± 38 µg/mL) or after (187 ± 32 µg/mL) spheroid formation. Finally, confocal microscopy highlighted a peripheral accumulation of QDs (4 h, 250 µg/mL) and a 36 µm penetration depth, before losing 50% of fluorescence intensity [102].

In another study, previous QDs (CdTe/CdS/ZnS) were coated with methoxy PEG thiol (mPEGthiol) to improve their colloidal stability and reduce their toxicity. mPEG coating was performed with different Zn/mPEGthiol molar ratio: (1:5) and (1:10). COOH-QDs (unpegylated) had a zeta potential of −36.8 mV, while mPEG-QDs (1:5) and mPEG-QDs (1:10) had a zeta potential of −31.1 and −25.9 mV, respectively. Compared to COOH-QDs, pegylation reduced QDs aggregation, nonspecific protein adsorption, and non-specific cell interactions, and these effects were more pronounced with mPEG-QDs (1:10) than that with (1:5). QDs toxicity was evaluated in both A549 monolayers and spheroids using Cell Tilter-Blue (CTB) assay after 24 h. In 2D, mPEG QDs were less toxic than COOH-QDs. In 3D spheroids, QDs were less toxic than in 2D, with still better biocompatibility for mPEG QDs (50% cell viability between 500–1000 µg/mL for mPEG QDs against 250 µg/mL for COOH-QDs). Morphological alteration of spheroids started with concentrations above 61 µg/mL for COOH-QDs and 125 µg/mL for mPEG QDs, leading to partial desegregation. Finally, confocal microscopy was performed on A549 spheroids. After 24 h, COOH-QDs and mPEG QDs (1:5) accumulate in the periphery and sometimes agglomerate, while mPEG QDs (1:10) have slower adsorption and more uniform binding. Thus, high density pegylated QDs were less subjected to nonspecific interactions and showed better biocompatibility [103].

### 5.2. QDs as a Part of Naniparticular Complexes

#### 5.2.1. Imaging Applications

Another way to address QDs to tumor cells is their encapsulation into liposomes. Al-Jamad et al. worked with pegylated QDs (−7.63 mV zeta potential, 40 nm diameter), and encapsulated them in small unilamellar lipid vesicles, resulting in the formation of 80–100 nm liposomes (QDs-L). Lipid composition was modulated to obtain low charged QDs-L (−7.86 mV to −16.7 mV zeta potential) or analogue of cholesterol was added to obtain cationic QDs-L (51.9 mV to 56.1 mV zeta potential). In 200 µm B16-F10 murine melanoma spheroids, 145 nM pegylated QDs could not label spheroid after 4 h, while cationic QDs-L labeled spheroids up to 30–50 µm depth. Zwitterionic QDs-L labeled spheroids deeper and more uniformly than cationic ones but with a weaker overall signal. Finally, syngenic B16-F10 tumors were implanted subcutaneously in C57B16 mice. Mice were injected in the tumor area with a dose of 33 pmol of QDs. A low signal was detected with pegylated-QDs or zwitterionic QDs-L at 5 min and 24 h post-injection while intense signal was detected with cationic QDs-L, even at 5 min. The QDs-L signal was concentrated in interstitial spaces and around cell membranes at 5 min, while it was localized inside tumor cells at 24 h. Both models confirmed that pegylated QDs could not label cells, notably due to steric hindrance, while encapsulation in liposomes improved delivery, especially for cationic liposomes [104].

#### 5.2.2. Imaging and Drug Delivery

Das et al. synthesized PLGA (Poly(D, L-lactide-co-glycolide)) nanoparticles encapsulating nutlin-3a and functionalized with EpCAM aptamer (Apt) and QDs 605 ITK^TM^ amino PEG for cancer theranostic purposes. Nutlin-3a acts as an anticancer agent, notably by stabilizing p53, while EpCAM was chosen as molecular target for cancer cells. Final NPs had a diameter of 292 nm and a −20.3 mV zeta potential. In monolayers, Apt-NPs induced higher labeling and cytotoxic effects in ZR751 breast cancer cells (EpCAM+) compared to Apt-free NPs, while no difference was observed in HEK-293 human embryonic kidney cells (EpCAM−). Then, ZR751 spheroids were generated using the liquid overlay method and were analyzed by z-stack confocal microscopy revealing a greater accumulation of Apt-NPs in spheroid center compared to Apt-free NPs after 4 h. These results confirmed the theranostic potential of these NPs, as conjugation with EpCAM Apt allowed QDs-assessed imaging (diagnostic) and nutlin-based therapy [105].

Another work conducted by Parhi and Sahoo focused on lipid nanoparticles of TPGS (D-α-tocopheryl polyethylene glycol succinate) encapsulating rapamycin and QDs 605 ITK^TM^ amino PEG (Rapa-NPs) and coated with the humanized anti-HER 2 antibody trastuzumab (Tmab-Rapa-NPs) for cancer imaging and therapy. This nanoformulation reached a diameter of 72 nm and a −11 mV zeta potential. These nanoformulations were tested in SKBR3 (HER2+) breast cancer spheroids. Better labeling was observed with Tmab-rapa-NPs compared to Rapa-NPs, and z-stack analysis of spheroids revealed a deeper penetration at spheroid center with Tmab-rapa-NPs; low signal was obtained with Rapa-NPs. Thus, these nanoformulations allowed both imaging and drug delivery [106].

Zhang et al. synthesized a nanomicelle platform consisting of TPGS and carbon dots and loaded with doxorubicin (DOX) to overcome cancer drug resistance. TPGS was modified with triphenylphosphine (TPP) to achieve mitochondrial targeting. TPP-NPs had a hydrodynamic diameter of 101.4 nm with a +21.0 mV zeta potential, while TPP-free NPs measured 87.6 nm, with a zeta potential of −10.8 mV. DOX release was higher in acidic conditions (65% at pH = 5, 18% at pH = 7.4), allowing specific release in tumor environment. The authors analyzed nanomicelles’ targeting ability in 300–400 µm spheroids. In MCF-7 spheroids, as incubation time increases up to 12 h, TPP nanomicelles penetrated deeper than DOX. Spheroid labeling from the periphery to center was observed up to 80 µm depth for DOX and 120 µm for TPP nanomicelles. Similar conclusions were set with MCF-7/ADR spheroids, except that only 2 h were needed to obtain a differential between DOX and TPP-nanomicelles. Finally, the authors showed a clear growth inhibition of MCF-7/ADR spheroids treated with TPP nanomicelles. After seven days, spheroids were smaller, disrupted, and fragmented while spheroids treated with free DOX were poorly altered. Thus, the suggested nanomicelles have succeeded in overcoming drug resistance of MCF-7/ADR cells in 3D models [107].

#### 5.2.3. PTT Combined with Chemotherapy

Wang et al. designed oxidized mesoporous carbon NPs encapsulating DOX and coated with CQDs to combine thermochemotherapy and infrared thermal imaging. CQDs were modified with tumor targeting moieties (CD_HA_) consisting of citric acid, hyaluronic acid polymers, and ethylenediamine to target CD44 receptors. MC-CD_HA_ had a strong absorption at 808 nm and a 27.4% photothermal conversion efficiency. The authors generated 200–300 µm 4T1 (mouse breast cancer) spheroids. After 24 h, spheroid size was reduced with free DOX, MC-CD_HA_ loaded with DOX, or PTT with MC-CD_HA_, and they were almost disintegrated when combining MC-CD_HA_ loaded with DOX with PTT. Finally, subcutaneous 4T1 tumors were generated in Balb/c mice and showed higher DOX accumulation with MC-CD_HA_ compared to free-DOX after 6 h. After 3 min of NIR irradiation, temperature around the tumor increased from 31.6 to 54.8 °C, allowing PTT and IR thermal imaging. Tumor growth was slowed down after each treatment but was inhibited and further reduced with MC-CD_HA_ loaded with DOX + PTT. Thus, the fabricated nanosystem efficiently combined PTT, chemotherapy, and IR thermal imaging in both models [108].

Within the same laboratory, Feng et al. used hollow mesoporous carbon (HMC) containing DOX, coated with ZnO QDs to combine PTT and chemotherapy in response to a triple stimulus: acidic pH, GSH, and NIR irradiation. Final HMC-ZnO had a 287 nm diameter, a −10 mV zeta potential, and was characterized by a 29.7% photothermal conversion efficiency under 808 nm NIR irradiation. The DOX release was higher with DOX/HMC than with DOX/HMC-ZnO, highlighting QDs implication to prevent early release of DOX, and was further improved after NIR irradiation. Cell viability was evaluated on 4T1 spheroids using Live/Dead assay. A total of 39% of spheroid cells were killed with DOX and mainly on the periphery, while DOX/HMC-ZnO or PTT alone (HMC-ZnO + NIR irradiation) killed around 60% of cells, deeper into spheroids. Combined therapy (DOX/HMC-ZnO + NIR irradiation) killed 87% of spheroid cells, from spheroid core to the periphery, resulting in spheroid dissociation. The study ended with in vivo mice bearing 4T1 tumors and showing higher tumor accumulation of DOX/HMC-ZnO after 6 h, allowing IR photothermal imaging of tumor and reducing tumor volume when combining PTT and chemotherapy [109].

Sung et al. synthesized a nanosponge (NS) enveloped by a red blood cell (RBC) membrane, conjugated to the anti-EGFR antibody cetuximab (Ct), and loaded with graphene QDs and docetaxel (GQD-D) for theranostic purposes. Final Ct-RBC@NS displayed a size of 260 ± 120 nm and a −10.2 mV zeta potential. Ct-RBC@GQD-D/NS had an efficient photothermal conversion, reaching 55 °C 5 min after NIR irradiation, which also triggered GQDs and docetaxel release. Then, 200 µm A549 spheroids were generated using microfluidic chips and NPs uptake was evaluated by fluorescence microscopy. RBC@NS remained in the media around spheroids while Ct-RBC@NS labeled spheroids surface. Furthermore, NIR irradiation (20 s, 1.5 W/cm^2^) increased NPs penetration and GQDs release up to the center of spheroids. After 24h incubation with 20 µg/mL of Ct-RBC@GQD-D/NS cell viability decreased until 30%, while dropping to 3% after 10 min of NIR irradiation. The study was finalized by injecting 1.5 mg/mL NPs in the tail vein of nude mice bearing A549 tumors. Fluorescence intensity was higher with Ct-RBC@NS compared to RBC@NS and GQDs could be found in tumor mass few hundred micrometers from vasculature after NIR irradiation. Biodistribution study revealed an accumulation of both NPs in the lungs and liver, although less than in tumors. After 10 min of NIR irradiation, tumors could be heated up to 68 °C with Ct-RBC@NS. A combination of chemotherapy and thermotherapy with Ct-RBC@GQD-D/NS or RBC@GQD-D/NS exhibited the best results in terms of tumor growth inhibition. Thereby, the proposed NPs efficiently combined chemo and thermotherapy, enhanced by effective drug delivery after NIR irradiation [110].

As summarized in Table 3, different types of studies have been carried out in which QDs have been incorporated to larger NPs. Among the proposed strategies, liposomes, mesoporous carbon, PLGA, TPGS, and nanosponges were employed. These approaches allowed encapsulation of QDs and thus improved their delivery, but also allowed to incorporate other molecules, such as chemotherapy agents or photothermal agents for PTT. Thereby, most of these NPs are part of the theranostic project, combining diagnosis and therapy. Similarly to QDs alone, the majority of these NPs had negative zeta potential, although their size was around one hundred nm while QDs did not exceed thirty nm after functionalization. Previously, the study of Ma et al. (in the Section 5.1.1) had shown that cationic QDs partially accumulated into spheroids while anionic QDs penetrated deeper [59]. Similar results were obtained in Al-Jamal et al. study where liposome encapsulating QDs penetrated deeper when their zeta potential was slightly negative, whereas cationic liposomes accumulated in the periphery. However, their in vivo murine melanoma models showed an opposite result where only cationic liposomes labeled tumors, likely through a better electrostatic interaction with tumors [104]. In other cases, NPs with negative zeta potential but functionalized for active targeting were used, which allowed consequent labeling of spheroids [105,106,107]. These studies used NPs to deliver both QDs and chemotherapy agents such as nutlin-3a, rapamycin, and doxorubicin to combine imaging and therapy. In addition to chemotherapy agents, photothermal agents for PTT have been encapsulated and delivered to spheroids to trigger their destruction under NIR irradiation [108,109,110]. Furthermore, NIR irradiation not only triggered PTT, but also enhanced QDs/chemotherapy agents release. In all three studies, combination of PTT and chemotherapy induced the most severe damage, leaving clusters of spheroids, with viability not exceeding 13%.

## 6. Conclusions and Perspectives

The optical properties of QDs and particularly their ability to emit fluorescence in the NIR range, as well as their resistance to photobleaching, make them ideal candidates for long-term dynamic imaging. Thereby, they could be used for fluorescence-guided surgery (FGS) to facilitate tumor resection and achieve negative surgical margins, thus improving a patient’s prognosis. Different types of QDs have been designed, differing by their chemical compositions, coatings, and tumor-delivery strategies. Nanocomplexes have been developed for theranostic purposes, combining the optical properties of QDs with other agents, in particular with those used for chemotherapy or phototherapy. Many studies have been carried out in vitro with 2D cell monolayers, but few were conducted in vitro in 3D models such as spheroids. Spheroids allow for studying QDs’ ability to target tumor cells in a 3D environment which mimics the constraints and challenges associated with small avascular tumor targeting (cell accessibility, heterogeneity, stratification, etc.). Penetration depth and QDs localization within spheroids are parameters which cannot be obtained in monolayers. These data allow the screening of QDs before moving towards in vivo models. Carbon-based QDs can be used as photosensitizers (PS) for PDT or photothermal agents (PA) for PTT, as well as nanocomplexes combining QDs with PS/PA. Tumor destruction through PDT/PTT can be simulated by spheroids, and notably help to evaluate if QDs penetration is sufficient to destroy the spheroid core, and if the hypoxic/anoxic area could hinder PDT efficiency.

Although a 3D environment can provide new data about QDs, spheroid models may also suffer from some flaws. Some of them, such as the lack of cellular diversity or an incomplete tumor microenvironment, can be overcome by forming spheroids in coculture, triculture, and more. Other models can be used, such as patient-derived organoids, providing patient-specific information and reconstructing tumor stroma. Microfluidic systems allow to precisely monitor culture parameters and to simulate vascularization. Bioprinting allows to reconstruct a 3D architecture comprising tumor cells and stromal cells within a scaffold recapitulating matrix composition. Nevertheless, spheroids remain a simple, low-cost, and reproducible model aiming to complete in vitro studies before moving to in vivo or clinical applications.

## Figures and Tables

**Figure 1 pharmaceutics-14-02136-f001:**
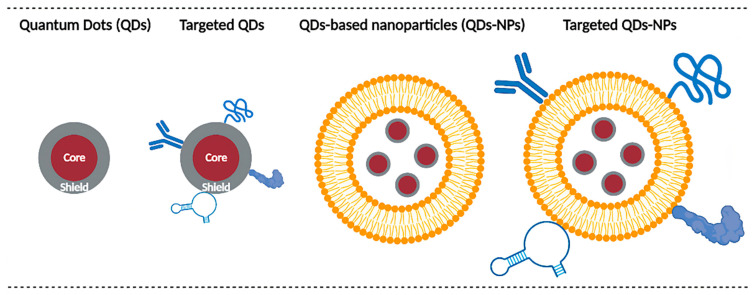
Main QDs delivery strategies. Delivery can rely on the intrinsic properties of QDs (size, charge, coating, etc.) or QDs can be functionalized with various ligands (e.g., antibodies, peptides, proteins, aptamers, etc.) for active targeting purposes. Another possibility is to encapsulate QDs in larger nanoparticles (e.g., micelles, liposomes, polymeric nanoparticles, etc.) to improve their biodistribution. These nanoparticles can themselves be modified for active targeting.

**Figure 2 pharmaceutics-14-02136-f002:**
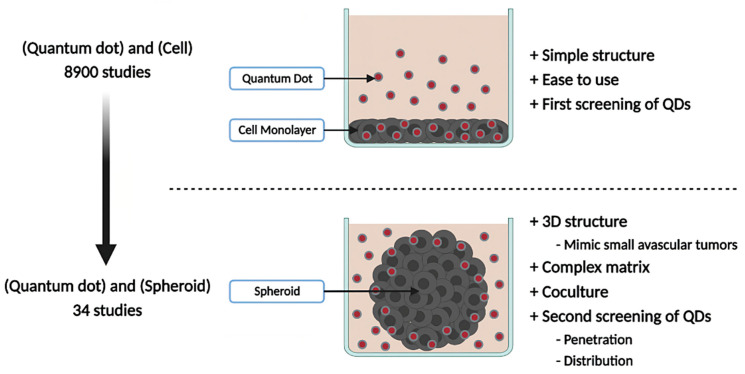
Advantages of QDs studies in 3D over 2D models. The majority of QDs studies are conducted on 2D cell models, having the advantage of being simple, easy to use, and necessary for the first screening of QDs candidates. 3D models such as spheroids allow to evaluate QDs behavior in small avascular tumors.

**Figure 3 pharmaceutics-14-02136-f003:**
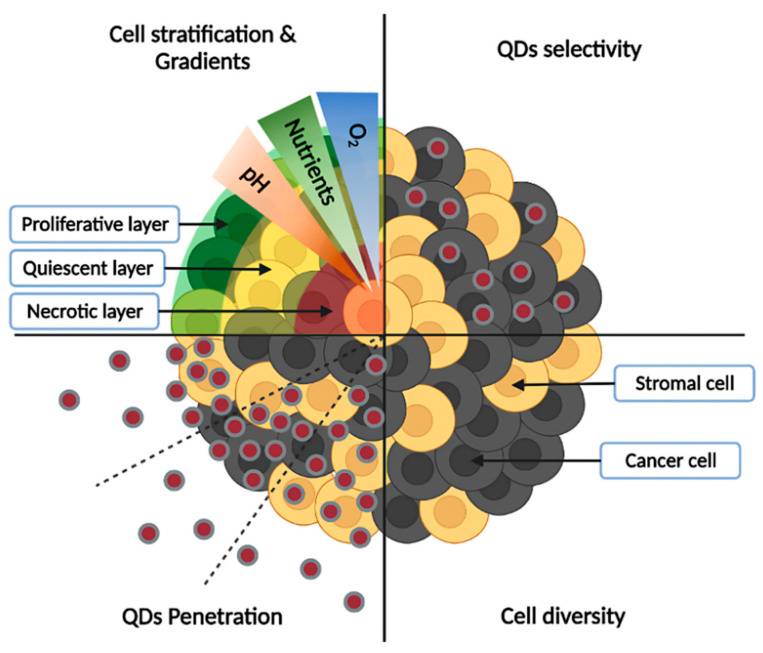
Characteristics of spheroid models and interest for QDs studies. Cell stratification into proliferative, quiescent, and necrotic layers, as well as the presence of gradients (low concentration of oxygen and nutrients in the core, accompanied by acidic pH) are depicted in the upper left panel, while cell diversity (presence of cancer and stromal cells) is illustrated in the lower-right panel. Spheroids allow the study of QDs penetration from the periphery to the core (lower-left panel), as well as their selectivity towards cancer cells (upper-right panel).

**Figure 4 pharmaceutics-14-02136-f004:**
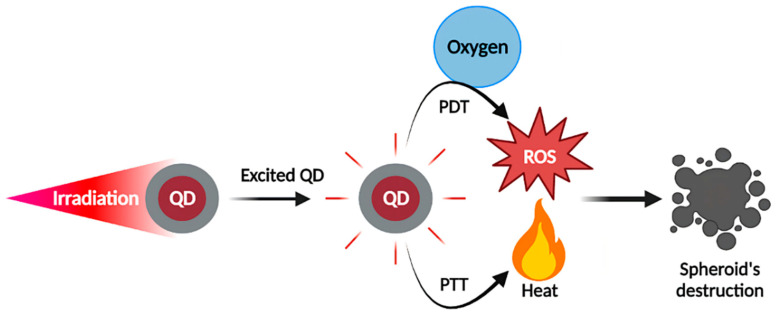
QDs as photosensitizers for photodynamic therapy or photothermal agents for photothermal therapy on spheroid models. Irradiation of QDs-loaded spheroids allows the production of ROS or heat (depending on the chemical properties of QDs) to mediate spheroid destruction by PDT or PTT, respectively.

**Figure 5 pharmaceutics-14-02136-f005:**
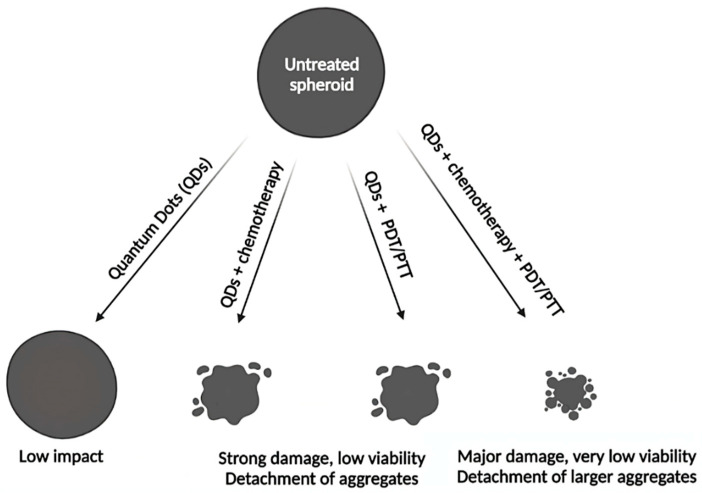
Simplistic representation of different treatment modalities induced by QDs aiming to destroy spheroids. QDs themselves have a low antitumor activity on spheroids. QDs combined with chemotherapy or used alone for PDT/PTT induced a significant decrease of spheroid size and viability, although not fully destructed them. This effect is further increased with combination of PDT/PTT with chemotherapy, demonstrating the highest cytotoxic effect. Please refer to experimental details [89,90,91,92] in the Table 2 of the main text.

**Table 1 pharmaceutics-14-02136-t001:** Main parameters and conclusions reported using untargeted and targeted QDs as imaging agents in spheroid models.

	Quantum Dots			Spheroids			
	QDs	Size (nm)	ζ Potential (mV)	Cells	Size (µm)	Conclusions	Ref.
Untargeted QDs	Qdot 625 ITK^TM^ carboxyl (CdSe/ZnS)	22	Negative charge	NIH3T3 MDA-MB-231 MCF-7	150–800 500 700	8 nM QDs labeled spheroid periphery up to 3rd–4th cell layer QDs penetration: 15 µm (1 h)–25 µm (24 h), independent of spheroid size MDA-MB-231: similar penetration profile to NIH3T3 MCF-7: deeper penetration due to spheroid looseness	[66]
	Qdot 605 ITK^TM^ amino PEG Qdot 605 ITK^TM^ carboxyl		Positive charge Negative charge	HeLa	540	No acute difference between QDs in monolayers (high labeling) Low penetration of positively charged PEG QDs in spheroid (20 nM) 90 µm penetration with negatively charged carboxyl QDs (20 nM)	[59]
	Qdot 655 ITK^TM^ carboxyl (CdSe/ZnS)	14.55	−35.1	MSCs MSCs/MDA-MB-231 MSCs/MCF-7	100 100 100	MSCs labeled with 8 nM QDs for 6 h before spheroid formation 31% of MDA-MB-231 were labeled after 24 h of spheroid coculture 18% of MCF-7 were labeled after 24 h of spheroid coculture QDs uptake in 3D vs. 2D: MDA-MB-231 (6-fold more), MCF-7 (1.5-fold less)	[67]
Targeted QDs	CuInS_2_/ZnS@MPA **+ iRGD peptide**	3.6	−21.5 −18.9	HeLa MSU1.1		Time-dependent labeling with 100 nM iRGD-QDs up to HeLa spheroid core. Labeling with unconjugated but not iRGD-QDs at 24 h in MSU1.1 spheroids	[68]
	Qdot^TM^ 605 streptavidin conjugate (CdSe/ZnS) **+ CPD**			HGM (HeLa)	350–400	10 nM CPD-QDs labeled whole spheroid after 6 h Thiol-mediated uptake allowed QDs penetration Spheroids were almost unstained with unconjugated QDs	[70]
	Qdot^TM^ 705 streptavidin conjugate (CdSe/ZnS) **+ anti-MCAM scFV**			Surgical specimens of mesotheliomas	150–200	50 nM scFV-QDs specifically labeled mesothelioma spheroid cells after 4 h No labeling with QDs conjugated to control-scFv	[71]
	QDot 545 ITK amino PEG + negative affibody QDot 605 ITK amino PEG **+ anti-HER2 affibody**	20.77 30.43 26.06 30.69		HCC1954 MCF-7 HCC1954/ MCF-7	120 120 120	Higher labeling with anti-HER2 QDs in HER2 + cells (HCC1954) Penetration up to 30 µm in HER2+ spheroids (up to 70 µm if permeabilized) In cocultured spheroids (HER2+/−): specific labeling of HER2+ cells Signal density was 9.2, 48.3, 30.8% in HER2−, HER2+, HER2+/− spheroids	[72]
	CdSe/CdS/ZnS coated with sulfobetaine **+ Folic acid (FA)**	18 18		KB	500	2-fold higher labeling with FA-QDs compared to QDs-free Saturation of FA receptor inhibited FA-QDs labeling Penetration depth: 100 µm	[73]
	CDs CDs-PEG-**FA**/DOX	2−9 12		BT549 T47D	120 125	CDs-PEG-FA/DOX injected in microfluidic device passed through vascular layer & ECM and reached spheroid Labeling of BT549 (FRα++) spheroid periphery at 12 h and core at 48 h Lower labeling of T47D (FRα+) spheroid periphery at 48 h Higher cytotoxic effect in BT549 spheroids	[74]
	CuInZnSe/ZnS coated with sulfobetaine **+ A20FMDV2 peptide (A20)**	26		FaDu FaDu/MeWo	427 478	Higher labeling with A20-QDs (25%) vs. QDs-free (3%) in both FaDu and FaDu/MeWo spheroids Higher labeling of FaDu cancer cells (32%) vs. MeWo fibroblasts (8%) A20 QDs penetrate up to 74 µm in monoculture and 56 µm in coculture	[32]
	Qdot 655 ITK^TM^ amino PEG (CdSe/ZnS) **+ PSMA**	33.4 34.2	−2.1 −1.3	LnCaP	1200 (450 depth)	Intense labeling after 2 h with 50 nM PSMA-QDs (61% of spheroid volume) Core and periphery were labeled, penetration up to 100 µm Low staining with QDs-free or saturation of PSMA receptor	[75]

Targeted moieties are highlighted in **bold**.

**Table 2 pharmaceutics-14-02136-t002:** Main parameters and conclusions reported using QDs as PDT/PTT agents in spheroid models.

Quantum Dots		Spheroids					
QDs	Size (nm)	Cells	Size (µm)	Phototherapy Parameters		Conclusions	Ref.
Carboxylated graphene QDs (+ treatment with DOX or 5-FU)	0.8	U87 PANC-1	400 400	PTT	808 nm, 5 min, 6 W cm^−2^	Most effective spheroid destruction with PTT (QDs) + DOX/5-FU Spheroid destruction occurred differently depending on cell type	[89]
Graphene QDs (GQDs) + 1.4% fluorine (F-GQDs)	3.7 2.1	HepG2	400–500	PDT	400–700 nm, 12 min, 40 mW cm^−2^	Brighter labeling with F-GQDs than GQDs PDT with F-GQDs drastically reduced spheroid diameter	[90]
Carbon QDs (CDs) Copper-doped CD (CU-CDs)	2.1 2.8	SH-SY5Y	300–400	PDT	400–700 nm, 12 min, 40 mW cm^−2^	Brighter labeling with CU-CDs than CDs PDT with CU-CDs drastically reduced spheroid diameter	[91]
Ag_2_S + 2MPA QDs + ALA (electrostatic) + ALA + **cetuximab (Cet)** + ALA + **Cet** + 5FU	5.3/13 * 3.5/19.5 * 5.9/49.8 * 6.3/114.4 *	SW480 HT29		PDT	420 nm, 5 min, 7 W cm^−2^	Higher QDs accumulation with cetuximab in EGFR+ cells Higher PpIX production with ALA-QDs and ALA-QDs + Cet SW480 viability after PDT: ALA (59%) < QD-ALA (47%) < QD-ALA-Cet (28%) < QD-ALA-Cet-5FU (18%) PDT efficiency in SW480 (high EGFR) > HT29 (low EGFR)	[92]

* Hydrodynamic diameter reported as number average and intensity average. Targeted moieties highlighted in **bold**.

**Table 3 pharmaceutics-14-02136-t003:** Main parameters and conclusions reported using QDs as part of nanoparticular complexes in spheroid models.

Nanoparticles (NPs)			Spheroids				
NPs	Size (nm)	ζ Potential (mV)	Cells	Size (µm)	Applications	Conclusions	Ref.
COOH-PEG-QDs QDs in cationic liposomes QDs in zwitterionic liposomes	40 80–100 80–100	−7.63 56.1; 51.9 −7.86; −16.7	B16-F10	200	Imaging	No labeling with COOH-PEG-QDs. 30–50 µm penetration with QDs in cationic liposome Weaker but deeper labeling with zwitterionic liposome. In vivo only cationic liposomes labeled tumors	[104]
PLGA encapsulating nutlin-3a PLGA + nutlin-3a, coated with **EpCAM aptamers** PLGA + nutlin-3a, coated with Qdot605 ITK^TM^ amino (PEG)	253 292 257	−16.5 −20.3 -8.9	ZR751		Imaging Drug delivery	Higher penetration & labeling in spheroid core with aptamers-NPs compared to aptamers-free	[105]
TPGS encapsulating rapamycin and Qdot 605 ITK^TM^ amino (PEG) **+ Trastuzumab**	72.6 72.0	−17.5 −11.0	KBR3	≈100	Imaging Drug delivery	Higher penetration & labeling in spheroid core with trastuzumab-conjugated NPs	[106]
TPGS nanomicelles + CQDs + DOX **+ TPP**	87.6 101.4	21.0 −10.8	MCF-7 MCF-7 /ADR	300–400	Imaging Drug delivery	TPP allows mitochondrial targeting of nanomicelles DOX loaded nanomicelles increased cytotoxicity in drug-resistant MCF-7/ADR spheroids Nanomicelles penetrate deeper (120 µm) than DOX (80 µm)	[107]
Mesoporous carbon (MC) NPs **+ CD_HA_** **+ CD_HA_** + DOX	≈230 ≈270	≈−35 ≈−28	4T1	200–300	PTT Chemotherapy Infrared thermal imaging	Allow PTT, chemotherapy, and IR thermal imaging Higher accumulation of MC-CD_HA_ in CD44 receptor overexpressing cells NIR irradiation increases DOX release Combined therapy induced spheroid destruction	[108]
Hollow mesoporous carbon (HMC) HMC + ZnO QDs	209 287	−39 −10	4T1	200–300	PTT, Chemotherapy, Infrared thermal imaging	Allow PTT, chemotherapy, and IR thermal imaging NIR irradiation increases DOX release Combined therapy induced spheroid destruction	[109]
Nanosponge (NS) **Cetuximab**-red blood cell membrane@GQD + docetaxel (Ct-RBC@GQD-D/NS)	170 ± 80 260 ± 120	−16.1 −10.2	A549	200	PTT, Chemotherapy Imaging	Spheroid uptake of Ct-RBC@NS but not RBC@NS Increased GQDs penetration after NIR irradiation Chemo and thermotherapy reduced spheroid viability to 3%	[110]

Targeted moieties highlighted in **bold**.

## Data Availability

Not applicable.

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
