# Peer review of "Quantum Dots Mediated Imaging and Phototherapy in Cancer Spheroid Models: State of the Art and Perspectives"

_pharmaceutics, 2022, doi:10.3390/pharmaceutics14102136_

Round 1

Reviewer 1 Report

Researchers Dirheimer et al. conducted a review of the applications of quantum dots in cancer imaging and therapy, with a particular emphasis on spheroids. The review is quite detailed but I have the following curiosity.

1.     The author should add a sentence or two to clarify the distinction between nanoparticles and quantum dots. Please explain how quantum dots are distinct from nanoparticles.

2.     Can 22 nm diameter size come under quantum dots? See Line 283. Does it fall inside the definition that is presented in Line 105 “QDs generally measure between 2 and 10 nm and have a spherical core”? Author, please make the necessary changes.

3.     Additionally, the author should add a heading for the many approaches that can be taken to synthesize/functionalize the QDs.

4.     I think the review does a good job of presenting the tables. On the other hand, some figures associated with the application are anticipated.

5.     The authors started the introduction with a discussion of the application of QDs to spheroids; however, as the discussion progressed, the authors shifted their attention to untargeted and targeted QDs in PDT and PTT, chemotherapy, and its toxicity. The monolayer format was utilized for the majority of the references and descriptions, as opposed to the spheroidal format. During the process of moving on with the document preparation, I got the impression that the authors strayed from the topic. It is possible that the author will decide to rearrange the document or rewrite the abstract/introduction as well as the title. The Abstract, Introduction, and title of the paper are written in a distinct style from the body of the text.

Overall, the manuscript needs major revision before it can be considered for publication in Pharmaceutics.

Round 2

Reviewer 1 Report

The author should provide the reference in figure 4 and figure 5 for the concept. For example, cite the paper which uses those concepts for therapy.
